# Implementing a hybrid cognitive-behavioural therapy for pain-related insomnia in primary care: lessons learnt from a mixed-methods feasibility study

Nicole K Y Tang [ORCID],[1] Corran Moore,[2] Helen Parsons,[3] Harbinder Kaur Sandhu,[3] Shilpa Patel [ORCID],[3] David R Ellard [ORCID],[3] Vivien P Nichols [ORCID],[3] Jason Madan,[3] Victoria Elizabeth Janet Collard,[1] Uma Sharma,[4] Martin Underwood[3,5]

[1]Department of Psychology, University of Warwick, Coventry, West Midlands, UK
[2]Department of Psychology, University of Leicester, Leicester, UK
[3]Clinical Trials Unit, University of Warwick, Warwick Medical School, Coventry, West Midlands, UK
[4]Patient Representative, Coventry, West Mindlands, UK
[5]University Hospitals of Coventry and Warwickshire, Coventry, West Midlands, UK

**Correspondence to**
Dr Nicole K Y Tang;
N.Tang@warwick.ac.uk

## ABSTRACT

**Objectives** To test the feasibility of implementing a brief but intensive hybrid cognitive behavioural therapy (Hybrid CBT) for pain-related insomnia.

**Design** Mixed-methods, with qualitative process evaluation on a two-arm randomised controlled feasibility trial.

**Setting** Primary care.

**Participants** Twenty-five adult patients with chronic pain and insomnia.

**Intervention** Hybrid CBT or self-help control intervention.

**Primary and secondary outcome measures** Primary outcomes measures were the Insomnia Severity Index and interference scale of the Brief Pain Inventory (BPI). Secondary outcomes measures were the present pain intensity rating from the BPI, Multidimensional Fatigue Inventory, Hospital Anxiety and Depression Scale and EQ-5D-5L.

**Results** Fourteen participants were randomised to receive Hybrid CBT, 11 to receive the self-help control treatment. Of the 14 in the Hybrid CBT group, 9 (64%) completed all four treatment sessions (4 discontinued due to poor health; 1 due to time constraints). Adherence to the self-help control treatment was not monitored. The total number of participants completing the 12-week and 24-week follow-ups were 12 (6 in each group; Hybrid CBT: 43%; self-help: 55%) and 10 (5 in each group; Hybrid CBT: 36%; self-help: 45%). Based on the data available, candidate outcome measures appeared to be sensitive to changes associated with interventions. Thematic analysis of pre-postintervention interview data revealed satisfaction with treatment content among those who completed the Hybrid CBT, whereas those in the self-help control treatment wanted more contact hours and therapist guidance. Other practical suggestions for improvement included shortening the duration of each treatment session, reducing the amount of assessment paperwork, and minimising the burden of sleep and pain monitoring.

**Conclusion** Important lessons were learnt with regard to the infrastructure required to achieve better patient adherence and retention. Based on the qualitative feedback provided by a subset of treatment completers, future trials should also consider lowering the intensity of treatment and streamlining the data collection procedure.

**Trial registration number** ISRCTN17294365.

### Strengths and limitations of this study

► First feasibility trial in the UK to evaluate a new, brief but intensive Hybrid CBT for pain-related insomnia compared with self-help control in primary care.

► The Hybrid CBT was manualised, the delivery of which was supported by a comprehensive therapist training programme.

► Patient recruitment was tested in three different health centres of different demographic compositions and different socioeconomic backgrounds.

► The mixed-methods approach provided both quantitative and qualitative information to inform the design and planning of a definitive trial.

► Rates of attrition and loss to follow-up were high in both arms.

## INTRODUCTION

Chronic pain is a major burden to primary care, accounting for 5 million general practitioner appointments each year in the UK.[1 2] These pain patients usually present with multiple symptoms, with insomnia being one of the most common and disruptive comorbidities.[2–4] In hospital pain clinics, as many as 90% of the patients report insomnia of a severity that warrants clinical attention.[3 5–8]

Conventionally sleep disturbance is seen as a secondary symptom to pain, but recent research has shown that poor sleep is actually a key driver for persistent pain and its associated distress and disability.[9–13] Additionally, untreated insomnia is a significant risk factor for adverse health outcomes, for example, hypertension, cardiovascular disease, obesity, diabetes, respiratory diseases and even increased mortality.[14–20]

Although better sleep has long been emphasised by pain patients as an important treatment outcome,[21 22] insomnia is rarely a focus in pain management programmes.

In primary care where most chronic pain patients are managed, hypnotics continue to be first-line treatments for insomnia despite the limited evidence supporting their long-term efficacy and safety.[23] Their prolonged use can result in undesirable side effects, increasing the risks of falls, road traffic accidents, dementia and mortality in the long term, particularly in older adults.[24–28] The risks multiply when the effect of polypharmacy is factored in. The combined use of benzodiazepines and opioids produce significant respiratory depression and is thought to contribute to the recent sharp rise in unintentional prescription drug overdose deaths.[29]

Psychological interventions offer a promising treatment alternative. The efficacy of cognitive behavioural therapy for primary insomnia (CBT-I) has been demonstrated by multiple systematic reviews and meta-analyses.[30–36] However, these treatments are often not available for chronic pain patients because of the lack of an empirically validated treatment protocol adapted and tailored for this population,[37] a shortage of skilled therapists[38–40] and an absence of essential infrastructure for CBT delivery in primary care.[28 30 41]

We have recently evaluated a talking therapy specifically modified for patients with pain-related insomnia.[42] The intervention simultaneously tackled chronic pain and insomnia, combining select components of CBT-I with interventions targeting the cognitive-behavioural processes maintaining chronic pain. The Hybrid CBT was delivered as an individual therapy over 4 weeks through weekly 2-hour sessions. The treatment dosage was 8 hours in total, approximating the optimal dose recommended for CBT for insomnia disorders[43] within a stepped care model.[39] In our pilot study with patients recruited from hospital pain clinics (ie, secondary care), the Hybrid CBT was associated with greater improvement in sleep at post-treatment compared with a symptom-monitoring control procedure (treatment effect size of Hybrid CBT: $d_H$=2.92; control: $d_C$=0.56).[42] Pain intensity did not change ($d_H$=−0.13; $d_C$=0.14), but the Hybrid CBT was associated with greater reductions in pain interference ($d_H$=1.92; $d_C$=1.19), fatigue ($d_H$=1.81; $d_C$=0.15) and depression ($d_H$=0.94; $d_C$=−0.04) than control.[42]

The current study tested the feasibility of adapting and implementing the Hybrid CBT in primary care, using a mixed-methods approach. With a small patient sample across three primary care centres from localities with different socio-economic and demographic characteristics, our overarching aim was to generate information to inform the development of a definitive randomised controlled trial (RCT) for evaluating the clinical outcomes and cost-effectiveness of the Hybrid CBT within the UK National Health Service setting. The focus of the current study was therefore not on detecting differences in outcomes between the Hybrid CBT and the control groups, but on evaluating the technical and logistic feasibility of a full-scale study. Specifically, our aims were to: (i) check participant's willingness to be randomised between the Hybrid CBT and self-help control intervention, (ii) assess recruitment strategies of practices, staff and patients, (iii) estimate attrition rates throughout the study, (iv) evaluate performance and acceptability of candidate outcome measures and (v) evaluate the data collection method. Hence, in this article, we report the methods and findings from the feasibility trial, along with qualitative findings based on our process evaluation of patient experience.

## METHODS
### Patient and public involvement
Two patient representatives with prior training for research involvement were recruited for this study through the Warwick Universities/User Teaching and Research Action Partnership. Our patient and public involvement representatives were co-applicants of the grant application and members of the project management committee. They were involved in most aspects of the study including trial design, therapist training, trial implementation and results discussion. Additionally, a member of our research team with a chronic pain condition gave significant insights into the running of the trial on top of their technical research expertise.

### Trial setting and design
The feasibility study was a RCT with a multicentre, parallel-group design situated within primary care. Treatments were offered to adults living with chronic pain and insomnia in the community and delivered by trained health psychologists at the primary care centre from which the patients were recruited.

Participants were randomly assigned to receive one of two trial treatments for pain-related insomnia, in addition to treatment as usual (TAU). Here, TAU referred to the existing advice and prescribed medications for pain and insomnia that the participants were receiving. The assumption of TAU reflected the clinical reality that most patients with these chronic conditions would have already received some medical advice or treatment for their symptoms.

### Treatments
#### Hybrid CBT
The Hybrid CBT comprised select components of CBT-I and interventions designed to target cognitive-behavioural processes maintaining chronic pain. The core components of the treatment were described by Tang et al[42] and included sleep psychoeducation, stimulus control therapy, sleep restriction therapy and cognitive therapy for addressing insomnia-related cognitions and behaviours common among patients with chronic pain. It also included individual formulation, goal setting and behavioural activation, components for reducing pain catastrophising and safety-seeking behaviour and reversing mental defeat for the management of chronic pain. The treatment was manualised for this study to facilitate therapist training; the guiding treatment principles

were laid out in the treatment manual to support flexible treatment delivery for this patient group with complex needs.

The format of the treatment can be described as 'brief, intensive and concentrated'.[44] Each patient allocated to the Hybrid CBT group was offered a total of four individual sessions on a weekly basis. Each session was approximately 2 hours long. The idea was to maintain the level of treatment content while minimising the burden of travel and duration of treatment, which may hinder treatment engagement in this patient group.[45]

### Self-help control treatment

Existing patient reading materials were amalgamated (with minimal modification) into four booklets to provide a self-help treatment on managing chronic pain and insomnia. The materials on insomnia were collated from the self-help treatment developed by Morgan et al.[46] Compared with TAU, the use of these self-help materials was found to be effective in improving insomnia symptoms in older adults attending primary care for sleep and other comorbid chronic conditions (post-treatment effect size on sleep measures ranged from d=0.69 to 0.7).[46] The self-help booklets were posted to the patients' homes, one at a time on a weekly basis. The content of the self-help gave equal coverage on chronic pain and insomnia management, approximating the structure and content of the Hybrid CBT. The self-help control treatment represented an active treatment control minus therapist contact.

### Therapists

In the UK, clinical psychologists are not usually a part of the primary care medical team. Patients are often referred to see a psychologist for psychological interventions on an as-needed basis. Other provisions of care exist, for example, in-house counsellors and local Improving Access to Psychological Therapy teams, but availability of these services varies depending on locality and resource allocation. Previous trials of CBT-I in primary care have recruited nurse/health visitors for the delivery of treatment.[47] Given the content (ie, treatment of two complex health conditions), approach (non-protocol-based CBT), format (individual, brief but intensive) and focus (behaviour change) of the Hybrid CBT, health psychologists were chosen as therapists for this trial.

Following targeted recruitment via health psychology training centres and professional networks across the UK, six health psychologists—fully qualified or in the latter stages of their stage II doctorate—were selected to receive 3 days of intensive training offered by the team. Three withdrew before the trial commenced due to clashes with existing employment/study commitments (eg, maintaining private practice and completing other training) and the distance of travelling involved in the process of treatment delivery (eg, from London or Staffordshire, where the recruited therapists were based, to Coventry, Rugby and Warwickshire, where the primary care centres were located). The remaining three (100% female) went on to become the trial's therapists, which involved further training at their own pace via learning resources posted online, case piloting, regular individual supervision by experienced health psychologists on the team (HKS, SP, NKYT) and travelling across sites to offer treatment to the patients in their localities.

### Patients

We recruited 25 people living with chronic pain, between April 2016 and April 2017 from three primary care centres in Coventry/Warwickshire of different demographic compositions (respectively having 2.1%, 3.7% and 25.8% non-white population) and different socioeconomic backgrounds (respectively scoring 1, 5 and 8 on the 1–10 Index of Multiple Deprivation).[48]

We identified participants for the study from the electronic registers held by the participating centres. Patients were initially screened by searching each centre's electronic patient records for inclusion criteria. We did two searches, with search terms broadened to include specific medications for the second search.

Inclusion criteria were individuals (i) aged 18 years or above, (ii) English-speaking, (iii) registered with one of the participating centres, (iv) with a history of chronic pain and insomnia (as indicated by their medical records), with (v) pain of at least moderate severity ($\geq$4/10 on a present pain intensity numerical rating scale for at least 6 months) and (vi) clinical insomnia ($\geq$15 on the Insomnia Severity Index (ISI), >3 nights a week, >1 month in duration). Criteria (vi) mapped onto the American Academy of Sleep Medicine Research Diagnostic Criteria for Insomnia Disorder,[49] which is consistent with the Diagnostic and Statistical Manual of Mental Disorders-5 diagnostic criteria, although the latter adopted a 3-month duration criteria.[50]

Participants taking pain/sleep medications on a stable regimen were included if they met criteria of the study. However, we excluded potential participants with diagnosed or suspected medical/psychiatric/sleep disorders for which CBT-I was contraindicated as first-line treatment or those who had recently enrolled in or were completing a pain management programme, or other psychological treatments for pain or sleep.

As this was a feasibility study, no formal power calculation to test the effectiveness of the intervention was possible. The current sample size was determined by practicalities and considerations that it was sufficiently large to ensure that randomisation was acceptable and to buffer against atypical attrition.

A randomisation list was created by the trial statistician (HP) using random blocks of varying sizes (block length=4 or 6). Blocks were generated in groups of patients at a 1:1 ratio, and stratified by centre from which the participants were recruited. Patients were randomised sequentially as they became eligible for inclusion in the study. Allocation was concealed using 'e-envelopes', which were macro-enabled MS Excel files preserved as read-only with automatic saving on any alteration to the

file to give an audit log. This method was considered cost-efficient for a small-scale feasibility study. Furthermore, the study statistician only released these e-envelopes and their passwords in small batches on request by the trial coordinator (CM). The study statistician had no contact with participants at any point in the study.

## Quantitative outcome measures

We piloted the use of five validated questionnaires to collect data in this population. Our candidate primary outcomes were the ISI[51] and Brief Pain Inventory (BPI) pain interference subscale.[52] Candidate secondary outcomes included the present pain intensity rating from the BPI,[52] Multidimensional Fatigue Inventory general fatigue score,[53] Hospital Anxiety and Depression Scale[54] and EuroQoL EQ-5D-5L,[55] for which health utilities were calculated using the UK tariff of the EQ-5D-5L value set.[56] The health thermometer score was also reported.

Several process measures were included as part of the assessment to inform treatment and elucidate the role of hypothesised treatment mechanisms. These were the Pain Catastrophizing Scale,[57] Pain Self Perception Scale,[58] Anxiety and Preoccupation about Sleep Questionnaire,[59] Dysfunctional Belief and Attitude about Sleep Scale-16[60] and Pain-specific Dysfunctional Belief and Attitude about Sleep Scale.[61]

In addition, the participants were asked to complete a daily sleep diary modified from the Consensus Sleep Diary[62] and to wear an actigraph (Model: MW8, supplied by CamNTech) for a week for baseline and 12-week follow-up assessments to examine the feasibility of incorporating objective sleep measures for future trials.

## Statistical analysis

As the aim of the current trial was to evaluate study feasibility, not treatment efficacy, no formal between-group analyses were planned. However, planned analyses consisted of the generation of descriptive statistics for all time points, across all participants as a group, and the Cronbach's alpha of the candidate primary outcome measures to assess if internal consistency of these measures was adequate when administered in a chronic pain patient sample with a minor modification (ie, the sleep item was removed from the BPI to avoid criterion contamination). The candidate primary outcome measures were also checked for correlation to establish if co-primary outcomes were needed for a definitive study.

## Qualitative participant interviews

Face-to-face semi-structured interviews were conducted before and after the intervention to explore participants' expectations of the intervention and their overall experience postintervention. These interviews were carried out individually with a subgroup (20%) of participants enrolled in the trial by a Research Fellow who specialises in qualitative health service research but was not involved in the treatment design and delivery process (VPN). The interviewer had no prior contact with the interviewees.

---

> **Box 1  Seed questions used in the semi-structured interviews pre-intervention and postintervention for both Hybrid CBT group and self-help control group arms.**
>
> A. Pre-intervention interview
> 1. We are very interested in your journey. When did you first notice your pain-related sleep disturbance?
> 2. Did you put it down to anything?
> 3. Did you talk to anyone about it?
> 4. What did you do to try to help?
> 5. Does anything make it better or worse?
> 6. Have you seen any health professionals or other practitioners?
> 7. What do you think is going to happen in this study?
> 8. What are you hoping to get from this study?
>
> B. Postintervention interview
> 1. How did you get on with the study?
> 2. Did you get any benefit from the study (attending the sessions/ receiving the booklets)?
> 3. What worked for you?
> 4. What did not work for you?
> 5. Would you recommend this type of programme to other people?
> 6. If we ran this programme in a larger study is there anything you would change?
> 7. How did you get on with the paperwork?
> 8. How did you find the process of being in a research study?

Seed questions used to prompt these conversations are shown in box 1.

Timelines of the participant's symptom and treatment journey were drawn during the interview and field notes were written immediately postinterview to promote reflexivity. The interviews were held at the participant's primary care centre, and lasted for about 60 min each. Interviews were audio-recorded on an encrypted digital device. Recordings were transcribed verbatim and anonymised. Data transcription was supported by another doctoral-level Research Fellow with experience in qualitative healthcare research but again not involved in the study design and treatment delivery process (VEJC). NVivo software was used for managing the data for analysis of the interview transcripts.

Thematic analysis was performed on the transcripts by the principal investigator (NKYT) following the six key processes recommended by Braun and Clarke[63]; familiarisation with the data set, generating initial codes, searching for themes, reviewing themes, defining and naming themes and report production. Codes and themes extracted were then reviewed by the interviewer (VPN) and a second experienced qualitative researcher on the team (DRE) to check for accuracy. Comments received were then used to revise the analysis. Themes were extracted with the awareness that questions asked in the pre-intervention interviews were different from the postintervention interview, and hence the analysis avoided referring to any within-participant change across these interviews. Constant comparisons were applied when analysing the postintervention interviews, contrasting the experience of the participants assigned to

receive Hybrid CBT with those assigned to the self-help control treatment.

## RESULTS

### Search results

The searches identified a total of 1434 potentially eligible patients across the three primary care centres (9.8% of all records). After receiving an invitation to become part of the study, 85 patients responded and were invited to telephone (n=45) and then in-person (n=40) screening to be further assessed for eligibility. Of the 25 (55.6% of screened) participants who were found to be eligible, all were recruited into the study. This corresponded to a pick-up rate of 1.7 per 1000 registered patients and a successful recruitment rate of 1.9 participants per month, or 0.6 participants per centre per month. Figure 1 specifies the number and reasons for exclusion at screening and subsequent stages.

### Baseline participant characteristics

Table 1 summarises the sociodemographic details provided by the study participants at baseline (n=25). At the group level, 56% of the participants were female with an average age of 49 years and a mean body mass index of 29. The majority of the participants were white (96%) and married or living as married (60%). Forty-four per cent of the participants had secondary level education as their highest educational qualification. At the time of the study, 40% of the participants were in paid work while 60% were unemployed, retired or engaged in other forms of activity, with 36% receiving some form of social benefits.

The mean reported duration of pain was 11 years. Most of the participants described their pain as 'constant' and the median number of pain sites reported was 5, with lower back as the most commonly identified pain site, followed by neck, shoulders, joints, legs, knees, arms, upper back, head and abdomen. The baseline present pain intensity VAS was 6.1, pain interference score was 6.6 and ISI was 18.9.

### Treatment adherence and attrition

All 14 participants assigned to the Hybrid CBT group completed session 1, 9 completed (64.3%) sessions 2, 3 and 4. Of the five participants who discontinued, four gave 'poor' health and one gave 'lack of time' as reasons for withdrawal (figure 1). Those withdrawn were recruited from primary care centres with more severe deprivation indices. For the self-help control group, all leaflets were mailed to the participants weekly, as per protocol. No further adherence data were collected. At 12 weeks, six participants were successfully followed up in each arm and at 24 weeks, five. This gave an overall participant retention rate of 48% at 12 weeks and 36% at 24 weeks.

### Outcome measures

Both primary outcome measures appeared to have excellent internal consistency. The Cronbach's alpha for the cohort was 0.94 (95% CI 0.92 to 0.97) for the ISI and 0.88 (95% CI 0.87 to 0.95) for the pain interference score. Even though the BPI sleep item was dropped to avoid criterion contamination, there was a strong non-zero correlation between ISI and BPI interference score (r=0.81, 95% CI 0.67 to 0.89).

Descriptive statistics of all primary and secondary outcome measures, as well as process measures, by assessment time points, are shown in table 2. No adverse events were reported in either allocation groups.

### Participant interviews

Five patients were interviewed; three were from the Hybrid CBT group and two were from the self-help control group. All completed both pre-intervention and postintervention interviews, except one in the Hybrid CBT group who only completed the pre-intervention interview.

#### Pre-intervention

Discussions in the pre-intervention interviews gave rise to six interesting themes, capturing several prominent psychosocial characteristics of the participants enrolled in the feasibility study. These themes (presented in table 3 with additional illustrative quotes) were concerned with the participants' sense of identity, personal adversities, treatment experience and coping strategies, perceived pain-sleep relationship, satisfaction with current service and treatment expectations.

### Pain changed who I am

There was a sense of damaged identity shared across the five participants interviewed. They appeared to define themselves by their experience of pain and losses due to pain. Frequent comparisons were made of what life was in the past, with what life is now and what life should be like at a certain age. The way in which participants spoke of their struggle with pain carried a sense of mental defeat.

> I've been suffering with sleep deprivation for many years…pain the same. … it destroyed everything really… life changed a hell of a lot, … it stole my life away. (Patient C)

### Pain and sleep did not occur in psychosocial vacuum

The interviewees' descriptions of their treatment journeys revealed that the issues of chronic pain and insomnia were embedded within a larger context of personal adversities. It was difficult to tell whether these were psychosocial triggers or consequences of chronic pain and insomnia. Specifics of these adversities revealed themselves at different places of the interviews, where details of personal lives were volunteered to situate the conversation. Adversities came not in isolation but in clusters as chronic pain and insomnia became increasingly more severe and disabling. Example adversities cited included

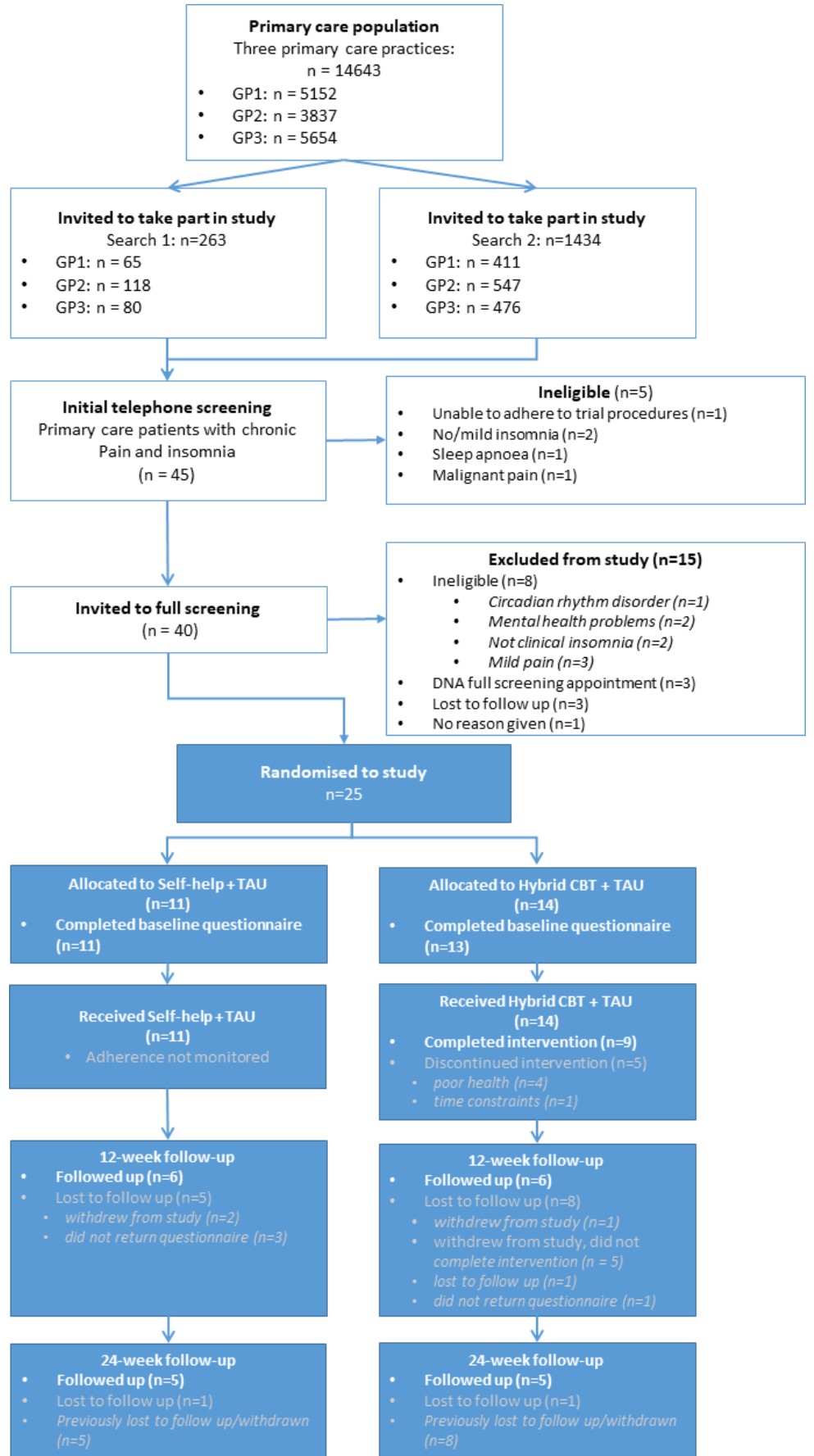

**Figure 1** Consolidated Standards of Reporting Trials flow diagram from screening (open boxes) to postscreening (filled boxes) processes in the study. GP, general practitioner; TAU, treatment as usual.

**Table 1** Participants' characteristics as measured at baseline

| Baseline variable | | All n=25 |
|---|---|---|
| Recruitment centre (n, %) | Primary care centre 1 | 8 (32) |
| | Primary care centre 2 | 10 (40) |
| | Primary care centre 3 | 7 (28) |
| Gender (n, %) | Female | 14 (56.0) |
| Age: years (mean, SD) | | 49.3 (9.8) * |
| Body mass index: kg/m$^2$ (mean, SD) | | 29.3 (7.6) |
| Ethnicity (n, %) | White | 24 (96) |
| Relationship status (n, %) | Cohabiting/Married/ Engaged | 15 (60) |
| | Single/Separating | 10 (40) |
| Education (n, %) | No formal qualifications | 8 (32) |
| | Secondary | 11 (44) |
| | Degree/Professional qualification | 6 (24) |
| Employment (n, %) | Paid work | 10 (40) |
| | Retired/Medically retired | 6 (24) |
| | Unemployed | 5 (20) |
| | Other | 4 (16) |
| Receiving benefits? (n, %) | Yes | 9 (36)* |
| How long have you had the pain? (minimum, in years) (mean, SD) | | 11.0 (9.0)† |
| What the pain is like? (n, %) | Constant | 20 (80) |
| | Recurrent | 3 (12) |
| | Occasional | 1 (4) |
| | Missing | 1 (4) |
| No. of painful places given (median, range) | | 5 (1–8) |
| Where is the pain? (n, %) | Head | 3 (12) |
| | Neck | 17 (68) |
| | Shoulders | 18 (68) |
| | Upper back | 6 (24) |
| | Lower back | 18 (72) |
| | Arms | 9 (36) |
| | Legs | 15 (60) |
| | Knees | 13 (52) |
| | Abdomen | 2 (8) |
| | Joints | 15 (60) |
| | Other | 7 (28) |
| Brief Pain Inventory (mean, SD) | Current pain severity | 6.1 (1.5) |
| | Current pain interference | 6.6 (1.5) |

Continued

**Table 1** Continued

| Baseline variable | All n=25 |
|---|---|
| Insomnia Severity Index Total score (mean, SD) | 20.1 (4.9) |

*One participant missing data.
†Three participants missing data.

ill health, mental health problems, car or work accidents, assaults, relationship breakdown, problems experienced by dependents or close family members, being a carer, job redundancy/unemployment, financial difficulties, homelessness and bereavement.

### Participants were not treatment naïve

Perhaps an artefact of self-selection bias in an RCT, all participants interviewed had tried to manage their pain and insomnia using a combination of drug and non-drug strategies, invariably with limited success. The self-helping spirit is a double-edge sword. If the right treatment is identified, it could facilitate engagement and maximise treatment gains. It could also drive people to take bold steps to keep pain and sleep problems under control, using strategies that are not necessarily recommended by current evidence-based guidelines, which may lead to dashed hope and further demoralisation.

> …Now I try to keep (pain medication) in my system all the time so there's always something there …rather than waiting for the pain to start and then taking (the medication). [Interviewer: Right so have you done that on the advice of somebody or off your own back?] No just off my own back… (Patient B)

### Pain was thought to be the primary cause of sleep problems

All participants interviewed shared a strong belief that pain was a major cause of their sleep problems. However, there was the awareness that other factors might also play a role in aggravating the sleep problems and that not sleeping well could have a reciprocal effect on the pain.

> [It's the pain that] keeps me awake…As soon as you have pain then wakes…Sometimes you can't sleep because of the pain, and then you're up all night, you get yourself angry with yourself… (Patient C)

### Participants were dissatisfied with the services available

All participants interviewed have had much interaction with multiple health service providers. They were most frustrated when they felt they were not being listened to or misunderstood by their GPs. They were also not happy with a lack of effective treatment choices, noting a contradictory combination of a heavy reliance on drugs for pain control with a general reluctance to prescribe sleeping tablets for insomnia.

**Table 2** Descriptive statistics of the candidate primary and secondary outcome measures, as well as process measures

| | Baseline | | 12 weeks | | 24 weeks | |
|---|---|---|---|---|---|---|
| | All (n=25) | No. of valid response (n, %) | All (n=25) | No. of valid response (n, %) | All (n=25) | No. of valid response (n, %) |
| **Primary outcomes** | | | | | | |
| ISI (mean, SD) | 20.2 (4.7) | 24 (96) | 14.4 (10.3) | 10 (40) | 14.8 (11.8) | 8 (32) |
| BPI interference | 6.1 (1.7) | 24 (96) | 4.9 (2.5) | 12 (48) | 4.7 (2.7) | 8 (32) |
| **Secondary outcomes** | | | | | | |
| BPI—pain intensity | 6.2 (1.6) | 24 (96) | 5.1 (2.4) | 12 (48) | 5.5 (2.6) | 8 (32) |
| MFI—general fatigue | 16.3 (3.6) | 24 (96) | 13.8 (3.7) | 12 (48) | 14.9 (3.2) | 8 (32) |
| HADS—anxiety | 9.7 (3.0) | 24 (96) | 6.7 (5.5) | 12 (48) | 8.9 (4.9) | 8 (32) |
| HADS—depression | 8.4 (3.5) | 24 (96) | 7.3 (4.6) | 12 (48) | 7.8 (5.1) | 8 (32) |
| EQ-5D—health thermometer score | 50 (17.5) | 24 (96) | 58.9 (15.2) | 12 (48) | 58.8 (23.0) | 8 (32) |
| EQ-5D—utility score | 0.60 (0.19) | 24 (96) | 0.57 (0.29) | 12 (48) | 0.56 (0.33) | 8 (32) |
| **Process measures** | | | | | | |
| PCS—pain catastrophising | 15.7 (9.2) | 24 (96) | 9.5 (8.5) | 12 (48) | 10 (8.6) | 8 (32) |
| PSPS—mental defeat | 30.8 (23.6) | 23 (92) | 21.9 (27.3) | 12 (48) | 14.7 (21) | 6 (24) |
| APSQ—sleep anxiety | 68 (21.7) | 24 (96) | 50.2 (32.4) | 12 (48) | 49.2 (37) | 9 (36) |
| DBAS—sleep beliefs | 5.6 (2.1) | 21 (84) | 4.1 (2.9) | 10 (40) | 4.0 (3.3) | 8 (32) |
| PBAS—pain-related sleep beliefs | 7.1 (2.1) | 21 (84) | 5.1 (3.4) | 12 (48) | 5.0 (3.7) | 9 (36) |

APSQ, Anxiety and Preoccupation about Sleep Questionnaire; BPI, Brief Pain Inventory; DBAS, Dysfunctional Beliefs and Attitudes and Sleep Scale; EQ-5D, EuroQol EQ-5D-5L; HADS, Hospital Anxiety and Depression Scale; ISI, Insomnia Severity Index; MFI, Multidimensional Fatigue Inventory; PBAS, Pain-related Dysfunctional Beliefs and Attitudes about Sleep Scale; PCS, Pain Catastrophizing Scale; PSPS, Pain Self Perception Scale.

I kept going with all these pain problems and that, never get nowhere… Just felt as though it was all in my head, nobody would listen to me… (Patient A)

### Participants' treatment expectations were high

All participants interviewed showed understanding that the interventions offered were not 'magic cures', but nonetheless had high expectations for the treatments, reflecting the constant tension between ideally what the patients want ('no pain' and 'lots of sleep') and realistically what can be offered by an intervention designed to optimise management of these problems.

Obviously it's not going to be magical overnight but even if it could give me ideas if I'm in that position how I should be reacting …. Well my future…hopefully I will have no pain … lots of sleep and no stress. (Patient D)

These themes, together, contextualised the feasibility study, offering finer insights into the life circumstances of those who actually signed up for pain-related insomnia treatment.

### Postintervention

Discussions in postintervention interviews revealed aspects of the intervention liked and disliked by the two interviewees (as graphically summarised for the Hybrid CBT in figure 2). Analysis of these factors was carried out separately for the Hybrid CBT and the self-help control groups, to generate clear suggestions as to how each of these interventions could be tweaked and improved from the patient experience perspective.

### Hybrid CBT

Besides 'very good practical advice' on sleep, both interviewees who completed the Hybrid CBT appeared to most appreciate the intervention for giving them a new understanding of sleep as well as themselves. In particular, they seemed to like the fact that these new insights enabled them to improve their sleep patterns and to change the way they think about and react to situations in life. Their success in improving sleep also appeared to have boosted their confidence in initiating changes in other areas of their lives; a spill-over therapeutic effect.

… You know, understanding how you sleep really… a lot better frame of mind now and a lot better myself

**Table 3** A summary of findings from the pre-intervention interviews, with additional quotes and implications for future trial planning

| Theme | Additional example quotes and/or notes* | Implications for future trial planning |
|---|---|---|
| Pain changed who I am | *"…you're in pain the whole time… (you) can't move and somebody's got to help you out of bed, which really at 48 I shouldn't be like that. … I feel I'm always going to be a person with pain…"* (Patient D)<br>*"I like to think that I'm quite strong but equally I feel that…I've given up. I'm frustrated (because) I'm hurting. I can't escape".* (Patient E) | Patients' damaged sense of identity—and the related psychological processes that feed into it—should be kept as a core target of the hybrid treatment and measured for pre-post intervention changes. |
| Pain and sleep did not occur in psychosocial vacuum | *No one single quote could satisfactorily illustrate the complexity of the psychosocial contexts described by the participants, and without risks of revealing their identities.\**<br>*Example adversities cited included ill health, mental health problems, car or work accidents, assaults, relationship breakdown, problems experienced by dependents or close family members, being a carer, job redundancy/ unemployment, financial difficulties, homelessness, and bereavement.* | While the current hybrid treatment has room to support flexible treatment delivery for patients with complex needs, more considerations should be given to the context in which the treatment is being delivered, as well as to practical support required to enable the most disadvantaged/ burdened patients to access treatment. |
| Participants were not treatment naïve | *"You just try to help yourself a little bit but, whether that's a good or bad thing I don't know".* (Patient A)<br>*"I'm trying to think myself healthy…I've tried … books … having your room right and spraying your pillow … all sorts of things…"* (Patient D)<br>*"…there wasn't nothing that I haven't already seen or read or something before…."* (Patient E) | Self-help treatments may not be considered as a satisfactory treatment option by this non-treatment naïve clinical population.<br>An active alternative treatment with therapist contact may be a more appropriate control intervention in future trials. |
| Pain was thought to be the primary cause of sleep problems | *"I'm not just gonna blame the pain…I've got a (teenage) son who's causing … I'm not naïve to think that's not a contributing factor (to sleep problems)… And I do stress … that's just in my nature".* (Patient D)<br>*"(when) I wasn't sleeping the pain seemed more unbearable… Unbearable, (because) I was tired… And I felt run down it just seemed worse I think…"* (Patient E) | If patients hold a rigid belief that sleep will never improve unless pain is resolved, it would be difficult to get their buy-in to the Hybrid CBT on offer. As such, these beliefs need to be addressed upfront in the information sheet or during recruitment, to improve treatment uptake and subsequent adherence. |
| Participants were dissatisfied with the services available | *"Most doctors these days don't…give the time of day. They've got your prescription written out before you go in".* (Patient C)<br>*"(Interviewer: So have you talked to anybody about your pain and sleep?) Only my GP…And they sort of got painkillers. They don't really like to give sleeping tablets anymore. Um… They advised over the counter ones…which work to an extent …"* (Patient B)<br>*"I don't feel that this surgery offers a lot of (non-drug treatments)…it can give (medication), but obviously I've stopped taking all tablets now for 5 weeks and I can't see if there's a difference from taking tablets to the placebo effect of fear that at least by reaching for the tablet there was something to help me".* (Patient E) | The issue of validation (or the lack thereof) is not unique to chronic pain patients, but highlights the importance for future trials to provide generic clinical skills training to the study therapists (health psychologists in the current study, or other suitably trained allied healthcare professionals with appropriate expertise in future trials). This will allow the provision of quality therapist contact, which is valued by our target patient group. |
| Participants' treatment expectations were high | *"…just to help control pain and sleeping…you can't work miracles but it might be something that can help me…To be honest … I'm hoping … you might have the magic cure, you never know".* (Patient A)<br>*"A bit more sleep. More than anything. I find if I'm tired, … the pain seems worse or I'm just not able to cope with it as well…So my theory is if I can just get a bit more sleep I can perhaps cope better with the pain".* (Patient B) | Proactive management of patients' treatment expectations at the outset of treatment, or as early as the enrolment stage in future trials, may help minimise attrition and unnecessary demoralisation. |

*Additional notes.
CBT, cognitive behavioural therapy.

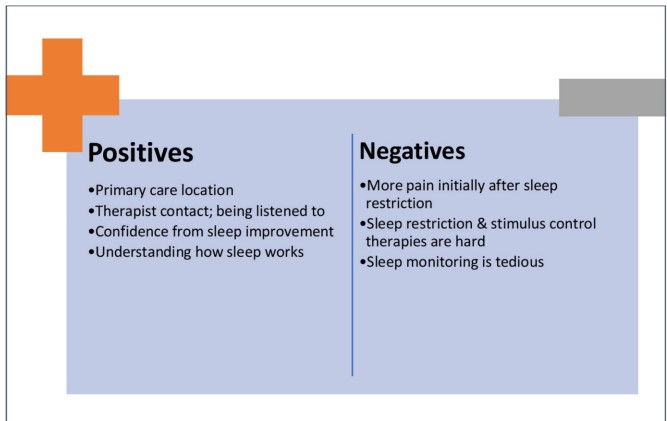

**Figure 2** A summary of themes from postintervention interviews of Hybrid CBT participants, highlighting the positives and negatives of the current treatment approach and content. CBT, cognitive behavioural therapy.

you know what I mean? …Because they're slowly dispersing these worries, and things like that, a lot to look forward to, and different things, more positive now… (It) changes the way you think a little bit and me talking to (the therapist) that I want to walk more and do more… It makes you think like… I can do something about it, I've done this, I've done um this sleep pattern and that… (Patient A)

I'm remembering everything and I really liked the study 'cause it helps me do things different… it's all about retrain(ing) your brain to you know understand our sleep patterns and things we do in our life, what we do and it works, it really really works… (Patient C)

Both participants interviewed were also very positive about the one-to-one, face-to-face interaction they had with the therapists. They felt that they were being listened to, and really appreciated the fact that the therapists went to the community to offer the intervention in their locality. They also felt that they managed to derive a greater understanding of the treatment materials because they could talk to someone and ask questions as they arose, compared with reading the information on their own. Furthermore, they felt that being able to talk confidentially to someone knowledgeable about their symptoms and experience had allowed them to process difficult emotions that may or may not be directly linked to the experience of chronic pain.

…the one-to-one sessions are good… They are really good. Um I don't know how it would work in a group. (Patient C)

…I think you understand it a bit more by talking (to) somebody (Patient A)

… I'd go home have a good read about [the treatment materials], some of it I didn't understand but then when I come back I'd ask… or she'd explain to me… write it down and show me that this is the way

to do it… So I was learning a new skill… I don't think I'd have took it so serious … I don't think I would've, just read a couple of leaflets and oh yeah and then popped it down the side, I wouldn't have thought about it… (Patient C)

… they could come to me and I didn't have to get there. I wouldn't have done that you see?… Cause it's on your door step it's a lot easier … and you fit it in (Patient A)

Three issues of the Hybrid CBT required attention. First, although sleep monitoring was an essential component of the intervention, the participants interviewed found completing the sleep diary tedious and possibly sleep-interfering as they felt obliged to clock their activities and remember everything they had done. Second, the participants found components of the sleep restriction and stimulus control therapy hard to follow. Although their sleep became more consolidated as a result of these treatment components, they did not particularly enjoy the experience and felt it was important to be able to personalise the intervention. Third, the participants struggled to apply the sleep restriction therapy during pain flare ups. Further adjustments to the pace and method of the therapy may be required considering that an increase in pain is a possible side effect during the initial stage of the sleep restriction therapy. Application of digital technology may also help reduce the burden of data collection.

[The monitoring/diary] was a bit monotonous, but then they're not going to know unless you write it down… (Patient A)

I didn't like that side of it (the sleep restriction and stimulus control therapy) 'cause I do like to stay in bed… (Patient C)

…but then you can have a flare up and then you're struggling…you know? (Patient A)

… I was following it all um you know we've come to like an agreement on timing to go to sleep and timing to get up…, but the problem is I've got a bad neck…It's really sore and moving it sometimes I can't move it … Can't go out, can't drive, can't do anything cause my necks playing me up, so it knocked my sleep pattern a bit out of proportion… (Patient C)

### Self-help control

The self-help control intervention was chosen to represent what was the best treatment option available for patients with chronic pain and insomnia in primary care. However, few positives were said about the self-help control intervention, except that the advice given was 'sound' and 'clear'. While both participants interviewed felt that overall the intervention was helpful, they felt that they were not learning much new information and could not pinpoint any specifics as to why and how the intervention was helpful. Their memories of what was being discussed in the self-help booklets were also very vague. They felt that the information provided in the booklets

overlapped with information available in existing self-help books or internet sources. They also indicated that they wanted more contacts with healthcare professionals rather than being left to their own devices.

> 'Yeah the study itself is alright…'… 'On the sleep (side), the things it gives are clear and can be helpful, but to me I already practiced those things anyway… but I've already read up on lots of things and done, so there wasn't a lot there that was explaining that I could take away and thought I've absorbed something new'. (Patient E)

> So something's obviously helped along the way but I can't put my finger on one exact thing. [Interviewer: is there anything, is there anyway we could improve it do you think?] In a way perhaps a bit more contact because obviously you have the initial study … I saw you for that interview sort of in the middle … And then now, but other than the booklets then you are just sort of left to it… There's no contact whatsoever… So some contact would've been useful. (Patient B)

Other feedback from participants assigned to both groups were concerned with the randomisation procedure, intervention format and data collection method. On randomisation, the participants in the self-help control group indicated that it 'would have been nice to have a choice', whereas those in the Hybrid CBT group imagined that they 'would not have bothered' with the treatment or would have benefited less had they been assigned to the self-help control group. There was also an agreement from participants of both groups on the intensity of the treatment, with too much to read and/or report. While they did not have an issue with automated data collection devices (ie, actigraphy), they had found it difficult to complete the sleep diary and recommended that the paper work of the treatment/study to be reduced.

## DISCUSSION

Delivering a brief but intensive intervention for the self-management of chronic pain and insomnia in primary care has proved to be challenging. Important lessons were learnt with regard to the infrastructure and trial design required to achieve better patient recruitment, treatment delivery, intervention adherence and patient retention.

### Were the patient identification and recruitment strategies viable?

Two searches were conducted to identify potential participants from electronic records. The initial search that used diagnoses as search terms identified only 263 potential participants across the three participating centres despite the high prevalence rates of both health conditions.[64–67] The unusually low return might be explained by the fact that physician records did not always list sleep or chronic pain as diagnoses. A second search using a broadened set of search terms (including medication prescribed) was

more successful, identifying 1434 potential participants (inclusive of the 263 previously identified). Of those who responded to the invitations and were invited to full screening, nearly a third were randomised into the study (63%). Of those who were not randomised, approximately equal numbers were ineligible and declined to attend full screening. This indicates that once found, recruiting patients into the study may not be an issue for a larger study, but capturing interested patients in the first place would be a challenge. While the current pick-up rate of 1.7 per 1000 patients was satisfactory, future trials should multiply the number of recruiting primary care centres to ensure that the recruitment target will be reached in a timely fashion.

Previous research has found that patients with insomnia tend to trivialise their symptoms and did not seek treatment due to beliefs that one should be able to cope with insomnia alone.[68] Despite the brief but intensive nature of the Hybrid CBT, 4 weekly sessions plus homework and data collection is a significant time commitment. Future trials should consider addressing these unhelpful beliefs during patient recruitment and incorporate an incentive system to motivate eligible participants to commit themselves to treatment. For example, a US trial of a 6 weekly 90 min group CBT for insomnia and pain among older adults with osteoarthritis offered a USD$2 cash incentive in the initial postal invitation.[69] Participants in this study were paid volunteers. They received a USD$50 incentive payment after completing the baseline assessment and attending the first class.[69] Similarly, an ongoing nurse-led brief insomnia treatment trial in the UK reimburses all participants after each completed follow-up visit; £5 at baseline, £10 at 3 months, £15 at 6 months and £10 at 12 months.[70] The current study offered the treatment for free but did not have the budget to incentivise enrolment and treatment attendance.

### What were the rates of attrition?

Loss to follow-up was high in this feasibility study, but return rates were generally higher in the Hybrid CBT group than the self-help control group. Adherence to the Hybrid CBT appeared to be most vulnerable after session 1. Ill health was cited as the main reason for drop-out and incidentally, all patients withdrawn were recruited from centres with more severe deprivation indices. Those who managed to return to session two fully adhered to the rest of the treatment programme and appeared to report sizeable improvements across outcome measures. Future trials should seek to investigate whether any systematic attrition from the Hybrid CBT occurs following session 1 compared with the control intervention, and if so, why? One size does not fit all; it would also be important for future trials to identify demographic, socioeconomic, and clinical factors that predict treatment suitability, directly answering the 'what works for whom' question.

In addition to a dedicated budget for incentivising follow-up rates, practical support (eg, travel reimbursement, appointment reminders, between-session technical

support) may also be required to remove participation barriers and reduce attrition in future trials. Qualitative feedback from our participant interviews also indicates that streamlining the data collection with the help of digital technologies or the use of a more active control intervention with more patient contact may help buffer against attrition. Although the Hybrid CBT was generally well received by treatment completers, it could be simplified in future trials to reduce patient burden. We note that the majority of the treatment drop-outs coincided with the introduction of sleep restriction component of the therapy in session 1 and prior to their return to session 2, when they were expected to report on their progress. Considering that sleep restriction is the most difficult and counterintuitive component of the therapy to follow, future trials with additional support for commencing sleep restriction may achieve better adherence to the intervention.

### How satisfactory and acceptable were the outcome measures and data collection methods?

Candidate outcome measures tested in the current trial appear to be psychometrically sound and have good face validity for the stated purpose of assessment. Both primary and secondary outcome measures showed changes in the direction anticipated for both arms of the trial over time. Care must, however, be taken when interpreting the positive responses that were reported due to biases associated with selective uptake and study attrition. We note that participants who remained in the study had lower levels of pain intensity at baseline than those who withdrew or were lost at the 12-week follow-up (BPI severity: 5.9 vs 6.4, not tested), although levels of pain interference and insomnia were more similar (BPI interference: 6.1 vs 6.2, ISI: 20.3 vs 20.1, not tested).

The pattern of change in the outcome measures was approximated by those of the process measures that assessed the hypothesised maintaining factors of pain and insomnia targeted by the treatments. Future trials should consider conducting appropriately designed analyses to examine whether changes in these processes mediate treatment outcomes. The combined use of a sleep diary and actigraphy is an important part of the assessment and treatment process (not reported in detail here), but the implementation of this recommended monitoring procedure proved to be challenging for the participants of the current study. Potential solutions to improve monitoring adherence in future trials may involve reducing the length of the monitoring procedure, more personalised data collection training and support and the installation of analysis software in participating centres to minimise human errors in data transfer.

### Strengths and limitations of this study

This was a small-scale feasibility study with a subgroup of participants being interviewed pre-post treatment for their experience participating in the trial. The specific aims were not focused on estimating treatment efficacy but on the implementation aspects of the running of the trial. We refrained from reporting the data by group or estimating effect sizes of treatment as the study was neither designed or powered to do so.[71 72] Bearing in mind the implementation issues discussed above, findings could be used to inform the design of any future definitive study but must be interpreted within its bounds of generalisability.

Therapist contact and the treatment focus on understanding how sleep works were highly valued by our interviewees. The delivery of the Hybrid CBT at the patient's doorstep was also positively received. Future trials should seek to maintain the quality of contact, although this will have to be supported by a robust health economic analysis that examines the cost-effectiveness of providing the Hybrid CBT in its current format as compared with in group settings or in the form of telemedicine or internet-based intervention.

While no adverse event was formally reported, qualitative feedback suggested that increased pain—especially in the initial treatment period—may be a possible side effect of the sleep restriction component of the treatment. Further investigations into the frequency, timing and severity of this potential side effect are required to ensure patient safety and, potentially, promote treatment adherence.

A previous meta-analysis has found the short-term to medium-term outcomes of patients with comorbid or primary insomnia receiving bibliotherapeutic self-help were superior to those of waitlist control.[73] Although self-help treatment is the best available non-pharmacological treatment options in many primary care settings, the trial team should consider whether it is an acceptable control intervention to offer in future trials. The post-treatment interviews revealed that our participants were not naive to self-help and appeared to be demoralised by the lack of therapist contact and fresh treatment content. Perhaps, a therapist-led educational intervention is a more fitting control for future trials evaluating the effectiveness of the Hybrid CBT in primary care.

### CONCLUSION

The Hybrid CBT has the potential to fill an unmet clinical need. Through our feasibility trial, a treatment protocol and a corresponding therapist training programme have been developed to make the delivery of this brief but intensive intervention in primary care possible. In its current form, the Hybrid CBT may work for subgroups of individuals who manage to adhere to the programme. Future trials could overcome the challenges highlighted in this feasibility study by broadening recruitment catchment, incorporating an incentive system to motivate treatment uptake, streamlining the treatment to make it even more primary care friendly and simplifying the data collection procedure to make it easier for the patients to take part and provide data for evaluation.

**Contributors** All authors have contributed to the interpretation of the data and manuscript preparation. NKYT was principal investigator and led the project. NKYT, HP, HKS, SP, DRE, JM, US and MU were involved in the design of the study. NKYT, HKS and SP led the treatment development and therapist training, with support from US. CM coordinated the recruitment of patients and data collection. VPN, DRE and VEJC conducted the qualitative interviews while HP conducted the quantitative data analysis.

**Funding** This paper presents independent research funded by the NIHR under its Research for Patient Benefit (RfPB) Programme (Grant Reference Number PB-PG-0213-30121).

**Disclaimer** The views expressed are those of the author(s) and not necessarily those of the NHS, the NIHR or the Department of Health and Social Care.

**Competing interests** All authors have completed the ICMJE uniform disclosure form. HKS and SP are directors of Health Psychology Services, which in part provides psychological treatments for those with chronic pain. HKS is chief investigator and co-investigator on other NIHR funded projects. MU was Chair of the NICE accreditation advisory committee until March 2017 for which he received a fee. He is chief investigator or co-investigator on multiple previous and current research grants from the UK National Institute for Health Research, Arthritis Research UK and is a co-investigator on grants funded by the Australian NHMRC. He is an NIHR Senior Investigator. He has received travel expenses for speaking at conferences from the professional organisations hosting the conferences. He is a director and shareholder of Clinvivo that provides electronic data collection for health services research. He is part of an academic partnership with Serco related to return to work initiatives. He is a co-investigator on a study receiving support in kind from Stryker. He has accepted honoraria for teaching/lecturing from CARTA & Sterling Events. He is an editor of the NIHR journal series, and a member of the NIHR Journal Editors Group, for which he receives a fee. NKYT received grant funding as co-investigator from NIHR for other projects and current funding from the Medical Research Council as principal investigator. NKYT is also a member of the trial steering committee of an ongoing trial funded by the NIHR.

**Patient consent for publication** Not required.

**Ethics approval** The National Health Service (NHS) ethical approval was given by the National Research Ethics Service (NRES) Committee West Midlands— Solihull, REC Number: 14/WM/1053. All participants gave informed consent before commencing the study.

**Provenance and peer review** Not commissioned; externally peer reviewed.

**Data availability statement** All data relevant to the study are included in the article.

**ORCID iDs**
Nicole K Y Tang http://orcid.org/0000-0001-7836-9965
Shilpa Patel http://orcid.org/0000-0003-0726-4888
David R Ellard http://orcid.org/0000-0002-2992-048X
Vivien P Nichols http://orcid.org/0000-0002-3372-1395

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
