## [Reviewer comments · BMJ Open]

ARTICLE DETAILS

TITLE (PROVISIONAL)	Implementing a hybrid cognitive-behavioural therapy for pain-related insomnia in primary care: Lessons learned from a mixed-methods feasibility study
AUTHORS	Tang, Nicole; Moore, Corran; Parsons, Helen; Sandhu, Harbinder; Patel, Shilpa; Ellard, David; Nichols, Vivien; Madan, Jason; Collard, Victoria; Sharma, Uma; Underwood, Martin

VERSION 1 - REVIEW

REVIEWER	Erin Koffel Minneapolis VA, USA The views expressed in this review are those of the author and do not reflect the official policy or position of the US Department of Veterans Affairs or the U.S. Government.
REVIEW RETURNED	12-Nov-2019

GENERAL COMMENTS	I enjoyed reading this manuscript and appreciate the multi-method feasibility approach used by the authors to prepare for a larger RCT. I have a few suggestions below. 1. Please clarify if the future trial will be a comparative effectiveness trial; that is, is the self help "control" condition a self-guided form of CBT-I that walks participants through the treatment or is it an educational control?2. It is unclear if the findings from participant interviews will be used to modify the future trial; although Table 3 has a column on "implications for treatment refinement and future trial planning" it is not stated if or how this information will be integrated into the future trial. Similarly, it is unclear how the negatives in Figure 2 will inform the future trial. It would be helpful to expand the discussion to include the implications of the qualitative findings as currently the discussion focuses on recruitment, attrition and data collection rather than treatment content.3. The "pre-intervention" subsection under participant interviews is very sparse; some more details on main themes and a few carefully chosen illustrative quotes would help readers navigate the table and have ready access to the main take away points.
--

REVIEWER	Michael V Vitiello, PhD University of Washington, Seattle WA USA
REVIEW RETURNED	08-Dec-2019

GENERAL COMMENTS	This well-written manuscript reports a mixed-methods feasibility study of the implementation of a hybrid cognitive-behavioral therapy for pain-related insomnia in primary care. Review of the manuscript raised some significant concerns, specifically:  1. A report and discussion of the provisional efficacy of the intervention would have greatly strengthened the manuscript. It is unclear why such was omitted. A focus on intervention feasibility in the absence of even preliminary evidence of efficacy is a weak and somewhat questionable exercise. Despite the author-noted, small sample size of the study some preliminary effect size estimates might be informative. 2. Tables 1 and 2 report the data for the two groups combined yet, given differential attrition across the study, it is important to know if the patients in the two arms differed across the variables reported. It would also address the efficacy concern raised above, as the combined group ISI and BPI scores reported suggest that there was considerable reduction in these scores in the active intervention arm. Such reporting of data for individual arms would also allow for judging if the self-help condition appeared to provide a good control, or if it too was efficacious. Optimally a good behavioral control would be expected to be associated with some benefit shown in outcomes, although nothing near that of an active intervention arm. This however is unlikely in this case given the following point. 3. The authors should address the major concerns that the self-help control was a very weak control in that it did not account for the intense patient/therapist interaction time (totaling 8 hours per patient) that was a major factor in the Hybrid CBT treatment condition. 4. The fact that only two patients, one from each treatment arm, participated in the post-treatment qualitative interview is a serious weakness. It is hard to accept that such an approach can extract meaningful "themes" of response. That is not to say that the comments of the post-test patients were not meaningful. The trenchant comment regarding self-help speaks directly to the concern with the design of the control detailed in point 3 above. Can the authors provide a rationale as to why less than 10 percent of post-test patients were so queried? This limitation needs to be addressed. 5. The abstract lacks a description of the lack of control dropouts at post treatment. It is left to the reviewer to assume that there were no dropouts. 6. Figure 2's balance scale theme, suggests that positives "outweighed" negatives and assumes that all factors are equally impactful, which may, or more likely may not, be the case. Suggest this Figure be changes to a simple list of the factors without the presumptive schema. 7. Table 3 is quite long. Some effort might be made to edit down patient quotes while maintaining their import.
---

VERSION 1 – AUTHOR RESPONSE

Reviewer: 1

Reviewer Name: Erin Koffel

Institution and Country:

Minneapolis VA, USA

I enjoyed reading this manuscript and appreciate the multi-method feasibility approach used by the authors to prepare for a larger RCT. I have a few suggestions below.

Thank you, Dr Koffel, for taking time to read the manuscript of this feasibility study and provide valuable comments. We appreciate it.

1. Please clarify if the future trial will be a comparative effectiveness trial; that is, is the self help "control" condition a self-guided form of CBT-I that walks participants through the treatment or is it an educational control?

Based on the findings of our current study, there appeared to be a need to tweak the content and delivery of the control intervention. In particular, there were suggestions from participants in the self-help control group that they wanted more therapist contact. A two-arm comparative effectiveness trial is a likely next step; a therapist –led educational control may help counteract demoralisation and attrition. We have now clarified this recommendation on page 28 of the manuscript.

2. It is unclear if the findings from participant interviews will be used to modify the future trial; although Table 3 has a column on "implications for treatment refinement and future trial planning" it is not stated if or how this information will be integrated into the future trial. Similarly, it is unclear how the negatives in Figure 2 will inform the future trial. It would be helpful to expand the discussion to include the implications of the qualitative findings as currently the discussion focuses on recruitment, attrition and data collection rather than treatment content.

As per suggested, we have now added a paragraph in the discussion (see pages 26 and 27) outlining the implications drawn from our experience running the current trial and the qualitative findings from patient interviews.

3. The "pre-intervention" subsection under participant interviews is very sparse; some more details on main themes and a few carefully chosen illustrative quotes would help readers navigate the table and have ready access to the main take away points.

We have now provided more details on pages 18-22 to summarise the main themes and to present a few illustrative quotes captured in these pre-intervention interviews. To avoid repetition, we have taken out the "Key points of the theme" column and the cited quotes from Table 3 (pages 19-20).

Reviewer: 2

Reviewer Name: Michael V Vitiello, PhD

Institution and Country:

University of Washington, Seattle WA

USA

This well-written manuscript reports a mixed-methods feasibility study of the implementation of a hybrid cognitive-behavioral therapy for pain-related insomnia in primary care.

Thank you, Dr Vitiello, for your helpful comments. Admittedly, some of the issues highlighted are limitations of the current feasibility study. As points 1 and 2 are meaningfully linked, our first response below addresses both points simultaneously.

Review of the manuscript raised some significant concerns, specifically:

1. A report and discussion of the provisional efficacy of the intervention would have greatly strengthened the manuscript. It is unclear why such was omitted. A focus on intervention feasibility in the absence of even preliminary evidence of efficacy is a weak and somewhat questionable exercise. Despite the author-noted, small sample size of the study some preliminary effect size estimates might be informative.

This is an important point raised. It remains our goal to evaluate the efficacy of the hybrid intervention in primary care in a future full trial, but for the current feasibility, our objectives were limited to testing the implementation aspects of the running of a trial of Hybrid CBT. Specifically, our aims were to (i) check participant's willingness to be randomised between the Hybrid CBT and self-help control intervention, (ii) assess recruitment strategies of practices, staff and patients, (iii) estimate attrition rates throughout the study, (iv) evaluate performance and acceptability of candidate outcome measures, and (v) evaluate the data collection method. These were stated in our funding application, research protocol and trial registration. Apologies for not communicating these clearly in the manuscript in the first place. We have now added these specific aims to the end of the introduction (see page 7).

As the current study was neither designed nor powered to detect differences in outcomes between the Hybrid CBT and self-help control groups, we did not carry out any statistical analysis to calculate treatment effect sizes (see page 12). Further, given the high attrition rates, we heed the recommendations by Thabane et al (2010) and Blatch-Johnes et al. (2018) and are cautious that reporting the results by group could be very misleading. We have now emphasised this as a limitation in the discussion (see page 28). Since both interventions evaluated in the current feasibility have had published data on their treatment efficacy, we have inserted this information in the introduction to help contextualise the findings (see pages 7 and 9).

References:

Thabane, L., Ma, J., Chu, R. et al. A tutorial on pilot studies: the what, why and how. *BMC Med Res Methodol* 10, 1 (2010) doi:10.1186/1471-2288-10-1

Blatch-Jones AJ, Pek W, Kirkpatrick E, et al
Role of feasibility and pilot studies in randomised controlled trials: a cross-sectional study *BMJ Open* 2018;8:e022233. doi: 10.1136/bmjopen-2018-022233

2. Tables 1 and 2 report the data for the two groups combined yet, given differential attrition across the study, it is important to know if the patients in the two arms differed across the variables reported.

It would also address the efficacy concern raised above, as the combined group ISI and BPI scores reported suggest that there was considerable reduction in these scores in the active intervention arm. Such reporting of data for individual arms would also allow for judging if the self-help condition

appeared to provide a good control, or if it too was efficacious. Optimally a good behavioral control would be expected to be associated with some benefit shown in outcomes, although nothing near that of an active intervention arm. This however is unlikely in this case given the following point.

Kindly please refer to our response above.

3. The authors should address the major concerns that the self-help control was a very weak control in that it did not account for the intense patient/therapist interaction time (totaling 8 hours per patient) that was a major factor in the Hybrid CBT treatment condition.

The self-help control intervention was based on materials previously used in a NIHR HTA-funded trial, evaluating a self-help bibliographical treatment for insomnia symptoms associated with chronic conditions in older adults attending primary care (Morgan et al., 2012). On top of that we added reading materials on chronic pain management that are freely available online via the websites of learned societies/organisations (e.g., NHS, British Pain Society, Pain Toolkit etc.).

In the Morgan et al. (2012) trial, the use of their self-help materials was found to be effective in improving insomnia symptoms compared to treatment as usual. Posttreatment effect sizes on sleep measures were medium for both Insomnia Severity Index ($d = 0.74$) and Pittsburgh Sleep Quality Index ($d = 0.69$). The effect sizes in these outcome measures attenuated but remained of similar magnitude at 3-month ($d = 0.51$; $d = 0.59$) and 6-month ($d = 0.56$; $d = 0.57$) follow-ups. These findings gave us the confidence that the self-help intervention would serve as a decent treatment-as-usual control, representing what the patients would have access to for the management of their insomnia and pain symptoms, in the absence of the Hybrid CBT. That said, we take the point that the self-help control did not control for therapist contact and future trials should consider adopting an enhanced controlled intervention with therapist contact. In fact, such a need to up the intensity of the control intervention was also flagged up by our qualitative findings (see page 27).

To clarify these issues in the manuscript, we have now provided more information about the self-help control intervention in the Methods section (see page 9). We have also discussed the drawback of the current self-help control in the limitation section (see pages 27-28) and recommend future trials to consider adopting an enhanced control intervention as a comparison to the Hybrid CBT.

Reference:

Morgan K, Gregory P, Tomeny M, David BM, Gascoigne C. Self- help treatment for insomnia symptoms associated with chronic conditions in older adults: A randomized controlled trial. *Journal of the American Geriatrics Society*. 2012;60(10):1803-10.

4. The fact that only two patients, one from each treatment arm, participated in the post-treatment qualitative interview is a serious weakness. It is hard to accept that such an approach can extract meaningful “themes” of response. That is not to say that the comments of the post-test patients were not meaningful. The trenchant comment regarding self-help speaks directly to the concern with the design of the control detailed in point 3 above. Can the authors provide a rationale as to why less than 10 percent of post-test patients were so queried? This limitation needs to be addressed.

The plan was to interview 20% of all participants (5 patients) pre- and postintervention to obtain qualitative information about themselves, their treatment expectations, and their treatment experience. Each interview was approximately 1 hour long. That should give 10 in-depth interviews for analysis but we had 9 as one participants did not give a post-intervention interview (see page 18).

The small sample size was a constraint of the time and resources we had. We named the insights drawn as “themes” given the rigorous extraction method employed as per the Braun & Clarke (2006) guidelines. However, we agree that the generalisability of the findings depends on how representative the interviewees were to the target population, as it is true for all qualitative studies. We have now further discussed the boundary of generalisation of these findings in the limitation and emphasized in the discussion that the themes extracted may not apply to treatment non-completers (see page 28).

5. The abstract lacks a description of the lack of control dropouts at post treatment. It is left to the reviewer to assume that there were no dropouts.

We have now specified in the abstract (page 4) that “Adherence to the self-help control intervention was not monitored”, to avoid confusion. We have also added a note in Figure 1 to clarify the issue.

6. Figure 2’s balance scale theme, suggests that positives “outweighed” negatives and assumes that all factors are equally impactful, which may, or more likely may not, be the case. Suggest this Figure be changes to a simple list of the factors without the presumptive schema.

This is a very good point. We have now replaced the figure with a simple list, to avoid making any implicit assumption of the weight of each theme. See the new Figure 2.

7. Table 3 is quite long. Some effort might be made to edit down patient quotes while maintaining their import.

As per suggested, we have now further edited the quotes to reduce the length of Table 3 from 6 pages to 2 pages.

Additional changes we have made:

We have now replaced the CONSORT checklist with CONSORT–P checklist, which is more suiting for our feasibility study. Given the changes made to the manuscript, we have updated the page numbers provided in the CORE-Q checklist.

VERSION 2 – REVIEW

REVIEWER	Erin Koffel Minneapolis VA, USA
REVIEW RETURNED	29-Jan-2020

GENERAL COMMENTS	The authors were very responsive to reviewer comments. I have no further concerns.
--

REVIEWER	Michael V Vitiello, PhD Department of Psychiatry and Behavioral Sciences University of Washington, Seattle, WA, USA
REVIEW RETURNED	29-Jan-2020

GENERAL COMMENTS	The revisions the authors have made in response to the comments of the reviewers have clarified and strengthened their already excellent manuscript. I have no remaining concerns.
--